# Multistage Detection of Tetrodotoxin Traces in *Diodon hystrix* Collected in El Salvador

**DOI:** 10.3390/toxins15070409

**Published:** 2023-06-25

**Authors:** Juan Carlos Fuentes-Monteverde, Marvin J. Núñez, Oscar Amaya-Monterosa, Morena L. Martínez, Jaime Rodríguez, Carlos Jiménez

**Affiliations:** 1CICA—Centro Interdisciplinar de Química e Bioloxía and Departamento de Química, Facultade de Ciencias, Universidade da Coruña, 15071 A Coruña, Spain; jufu@mpinat.mpg.de; 2NMR Based Structural Biology, MPI for Multidisciplinary Sciences, Am Fassberg 11, 37077 Göttingen, Germany; 3Laboratorio de Investigación en Productos Naturales, Facultad de Química y Farmacia, Universidad de El Salvador, San Salvador 01101, El Salvador; marvin.nunez@ues.edu.sv (M.J.N.); morena.martinez@ues.edu.sv (M.L.M.); 4Laboratorio de Toxinas Marinas, Escuela de Física, Facultad de Ciencias Naturales y Matemática, Universidad de El Salvador, San Salvador 01101, El Salvador; oscar.amaya@ues.edu.sv

**Keywords:** tetrodotoxin, HPLC-HRMS, HPLC-MS/MS-SRM, HPLC-HRMS^2^, *Diodon hystrix*, MZmine, natural products detection

## Abstract

This study describes a multistage methodology to detect minute amounts of tetrodotoxin in fishes, a plan that may be broadened to include other marine organisms. This methodology was applied to porcupinefish (*Diodon hystrix*) collected in Punta Chiquirín, El Salvador. A three-stage approach along with post-acquisition processing was employed, to wit: (a) Sample screening by selected reaction monitoring (HPLC-MS/MS-SRM) analyses to quickly identify possible toxin presence via a LC/MS/MS API 3200 system with a triple quadrupole; (b) HPLC-HRFTMS-full scan analyses using an ion trap-Orbitrap spectrometer combined with an MZmine 2-enhanced dereplication-like workflow to collect high-resolution mass spectra; and (c) HPLC-HRMS^2^ analyses. This is the first time tetrodotoxin has been reported in *D. hystrix* specimens collected in El Salvador.

## 1. Introduction

Tetrodotoxin (TTX) is a powerful neurotoxin with a molecular weight of 319 Da, which is widely distributed in nature [1]. Since the first report of this toxin isolated from the ovaries of a pufferfish by Tahara and Hirata at the beginning of the 20th century, TTX has been found in terrestrial vertebrates [2,3], invertebrates [4,5,6], bacteria [7,8,9,10], and fishes, including *Diodon hystrix* [11,12,13], among others [14]. Although this toxin has been found in several organs of adult pufferfish, TTX is mostly localized in the skin, muscles, intestines, gonads, kidneys, and liver [15]. The true origin of TTX remains controversial. Its production by symbiotic bacteria (endogenous route) or by accumulation through the diet are the most accepted proposals. Moreover, feeding on TTX-bearing microalgae by fish and its consequent accumulation is another explanation [9,16,17].

The deadliness of this potent neurotoxin is caused by its capacity to inhibit voltage-gated sodium channels and thus provoke death through respiratory paralysis [1,18]. The estimated minimum lethal dose value in humans is approximately 10,000 mouse units, nearly 2 mg [19]. The most useful approach to analyzing TTX is the LC/MS technique. A review of the advantages and limitations of this technique to detect and analyze the presence of TTX and its analogs has been recently reported [20].

The pufferfish *D. hystrix* (Linnaeus, 1758, Diodontidae), also known as spot-fin porcupinefish, is characterized by being one of the most predominant shell-crushing predators of mollusks in tropical coastal waters worldwide [21]. *D. hystrix* is distinguished for bearing spines on the caudal peduncle, fins with dark spots, and a broad head, as well as slight differences in the number of fin rays. *D. hystrix* is widespread in tropical and subtropical waters of the Indian, Atlantic, and Pacific Oceans. It also dwells in the circumtropical eastern Pacific from San Diego, California, to Chile, including the Galapagos Islands, and in the western Atlantic, from Massachusetts and Bermuda, continuing down to the northern waters of the Gulf of Mexico all the way down to Brazil [22,23].

The characteristic conditions of the high concentration of suspended material, coastal currents, and surface temperatures that range between 26 and 31 °C present in the tropical Pacific Ocean adjacent to El Salvador create a favorable environment for several tropical species. Although *D. hystrix* is spread throughout the entire coast of El Salvador, from the estuary of Rio Paz (13°44′39″ N, 90°07′58″ W) to Meanguera Island (13°12′4″ N, 97°41′42″ W), it is not commercially fished or consumed by Salvadorans, a distaste that is shared by others in Central America and the Caribbean.

Cases of fish poisoning in Hawaii (USA) [24] and Papua New Guinea [25] have been reported after *D. hystrix* consumption or following minor injuries caused by stabs from its spines in The Netherlands [26]. TTX occurrence, however, was not confirmed through analytical methods in the forenamed cases. Several studies focusing on TTX detection in *D. hystrix* have been reported. A study published in 1997 on the neurotoxic and cytotoxic activities of skin secretions from a specimen collected on the northern coast of Sao Paulo State (Brazil) suggested the presence of neurotoxins other than TTX, although the existence of a residual amount of TTX is not discarded [27]. As reported in 2014, the presence of TTX was detected and quantified by LC-MS/MS in the liver, but not in the muscle, of specimens from Sabah and Sarawak waters in Malaysia [11]. A genotoxicity study of TTX extracted from different organs of *D. hystrix* from the southeastern Indian coast, reported in 2016, displayed its presence in the eyes, liver, intestines, and gonads. In this study, GC-MS analyses confirmed TTX occurrence in all raw extracts except the eye. Extracts obtained from the skin, gonads, and liver caused 100% mortality when they were tested on Zebra fish [12]. More recently, TTX and analogs were detected and quantified by LC-MS /MS in the liver of four *D. hystrix* specimens collected on the southern coast of Mozambique [13]. In contrast, TTX was not detected by LC-MS studies in a *D. hystrix* specimen collected in Hawaiian waters nor in 42 distinct bacterial strains cultivated from this pufferfish [28]. In another study, no significant level of TTX was found in any organs of a *D. hystrix* specimen collected in Okinawan waters in Japan [29]. Additionally, TTX uptake into liver tissue slices of *D. hystrix* displayed no significant increase in TTX content over time during incubation. This finding suggests a limited ability of this toxin for bioaccumulation [30]. These studies seem to indicate that TTX presence in *D. hystrix* is associated with the geographic distribution of this pufferfish.

Continuing our search for bioactive natural products from different types of marine organisms [31,32,33], and given the lack of TTX studies of pufferfish from El Salvador, we endeavored to investigate the occurrence of TTX in *D. hystrix* collected in El Salvador due to the lack of TTX studies in pufferfish from these coasts. This paper reports the occurrence of TTX in extracts from different organs of *D. hystrix* sampled in Bahia Chiquirín, El Salvador, for the first time. A multistage methodology based on different MS techniques allowed the detection of TTX in minute amounts. These findings have given more insight into the worldwide distribution of TTX in *D. hystrix*.

## 2. Results and Discussion

Specimens of *D. hystrix* collected on the South coast of El Salvador were classified by gender (male, female, and undefined), and their organs, i.e., skin, liver, kidneys, flesh, and gonads, were dissected. All specimens collected were analyzed (three females, one male, and two undefined) (Table 1, Appendix A). Upon processing the samples (Appendix A), they underwent submission to HPLC-MS/MS-SRM, HPLC-HRFTMS, and HPLC-HRFTMS^2^ analysis. The main criteria were that the parent ion and at least one fragment ion of the toxin, as well as its retention time, be discernible [34]. Moreover, blank samples were included to rule out false positives.

### 2.1. HPLC-MS/MS-SRM

The first method to detect the toxin in extracts of *D. hystrix* implied the use of HPLC-MS/MS-SRM. It indicated the presence of the two characteristic parent-product ion transitions, 320/302 and 320/162, associated with TTX in the FG and MU samples (Figure 1 and Appendix A, respectively).

However, further studies of the mass spectra showed that even though the retention time variation (Δt*_R_* = 0.03 min) was within the accepted tolerance intervals in both FG and MU, the transition ion ratio (TIR) was outside the acceptable range (1.3 ± 0.2) in both samples, namely 0.59 and 0.21, respectively (Table 2). These findings indicate that the toxin concentration in FG and MU would be between the limit of detection (LOD: 72 ng/mL) and the limit of quantification (LOQ: 320 ng/mL). See the calculation of detection and quantification limits of HPLC-MS/MS-SRM in the SM (Appendix A) [35,36,37]. Thus, the concentration of TTX in those organs would be within the range of 4.1 and 18.1 μg.Kg^−1^, which corresponds to a value between 2 and 9 mouse units [13]. In order to corroborate the presence of the toxin, additional experimental approaches were employed, including high-resolution parent ion separation and two high-resolution stages.

### 2.2. HPLC-HRFTMS

HPLC-HRFTMS, operating in Selected Ion Monitoring (SIM) and positive ionization modes (ESI-(+)), was the second method used to detect TTX in *D. hystrix.* The method was corroborated by a standard solution, shown in Appendix A. The sample analyses evidenced the occurrence of TTX in both the female skin (FS) sample (Figure 2 and Appendix A) and the male liver (ML) sample.

The extracted Ion chromatogram (EIC) in a mass range of *m*/*z* 320.00–320.20 (Figure 2a) in the FS sample displayed a sharp peak with Δt*_R_* = 0.4 min. The exact mass spectrum was computed by averaging the high-resolution mass detected at three different retention times taken from the EIC (Figure 2b). The calculated mass measurement error (∆*m*/*z*) of FS 0.75 (0.62) ppm matched the TTX molecular formula (Figure 2c) [38,39,40]. Moreover, the sample EIC fully overlapped that of the standard sample (Appendix A). The Orbitrap detector was permitted to positively confirm the occurrence of TTX in the FS sample. The presence of TTX in the ML sample was also displayed by this method (Appendix A) based on the HRMS values (Δ*m*/*z* = 1.67 (0.5) ppm), although in this case, the Relative Isotope Abundance error (RIA*_error_*) was larger than the maximum tolerance value, presumably due to the low ion counts [41]. The Orbitrap detector has a lower threshold setting, below which no ions can be detected or accurately reported.

Once TTX was detected in the FS and ML samples by using the HPLC-HRFTMS method, we endeavored to find the most common TTX analogs (4,9-anhydro-TTX; 5-deoxy-TTX; 11-deoxy-TTX; 5,11-dideoxy-TT; 6,11-dideoxy-TTX; 5,6,11-trideoxy-TTX 11-nor-TTX-6(R)-ol; and 11-oxo-TTX) in those samples by resorting to that methodology [42,43]. As a result, no other analog of TTX was detected in either sample.

### 2.3. Post-Acquisition Data Analysis by MZmine

After the presence of TTX had been established by HPLC-HRFTMS in the *D. hystrix* samples, we provided a workflow allowing the automatization of toxin data analysis/dereplication. Laboratories wishing to automatize natural product detection should validate their data-analysis protocols. For data processing and metabolite detection, MZmine (V.2) has been extensively utilized in metabolomics and natural product dereplication, along with LTQ Orbitrap hybrid mass spectrometry [44,45,46,47,48,49]. The data analysis workflow software used here is shown in Section 4.

The automatized mode was employed to process and analyze the Orbitrap HPLC-HRFTMS files. After treating the FS sample raw file with Xcalibur^TM^, a peak attributed to TTX emerged at 8.1 min. The identification of TTX in the FS sample was achieved with the help of existing high-resolution MS, isotope pattern, and Ring/Double Bond Equivalent (DBE) available in online databases (Figure 3). We concluded that by using this scheme, it was possible to routinely automatize toxin detection and ease data processing.

### 2.4. HPLC-HRFTMS^2^

Toxin presence in *D. histrix* was corroborated by multiple-stage mass spectrometry in the LU sample. The method was corroborated, as before, by using a TTX standard solution (Appendix A). To optimize the signal-to-noise ratio, parameters were configured by utilizing a standard solution (see Section 4). The stock solution showed two distinct transitions, suggesting the appropriateness of this method for TTX detection (Appendix A).

The analyzed fragment ions found in LU displayed molecular formulas that matched those reported in the literature [34]. The parent ion at *m*/*z* 320.1088 (C_11_H_18_N_3_O_8_^+^) yielded distinctive fragment ions at the following *m*/*z* values: 302.0983 (C_11_H_16_N_3_O_7_^+^, Δ*m*/*z* = 0.03 ppm) and 162.0656 Da (C_8_H_8_N_3_O^+^, ∆*m*/*z* = 3.64 ppm). The Ring/Double Bond Equivalent (DBE) value changed from 4.5 to 5.5 for the fragment at *m*/*z* 302.0983, suggesting the loss of a water molecule accompanied by an adjustment on the ring system and resulting in the formation of an extra 5-member ring [9-O-4-4a-8a] (Figure 4a) [40]. On the other hand, the DBE value change from 5.5 to 6.5 for the fragment at *m*/*z* 162.0662 implied a significant modification in the molecular structure, suggesting the break of several bonds and the ensuing formation of a heterocyclic aromatic structure (Figure 4b). Therefore, based on the experimental evidence obtained through HPLC-HRFTMS^2^ experiments, the presence of TTX in the LU sample was also confirmed.

## 3. Conclusions

A trace detection methodology by MS techniques was proposed and applied to detect TTX in specimens of porcupinefish (*D. hystrix*) collected in Punta Chiquirín, El Salvador. The same extraction scheme for the toxin was used in all the samples, and an extra purification step was included in the HPLC-HRMS^2^ method to avoid matrix interference. Several techniques, such as HPLC-MS/MS-SRM, HPLC-HRFTMS, and HPLC-HRFTMS^2,^ were employed in conjunction with post-processing data using MZmine 2. The approach outlined here can be adjusted to study the occurrence of TTX in various other marine organisms with an economic interest.

The toxin was detected in the gonads (FG) and skin (FS) of the female specimens, in the liver of the male (ML) specimen, and in the liver (LU) and muscles (MU) of the undefined specimens (Table 2). The quantification of TTX could not be determined in any of the samples, most likely due to toxin levels falling below the limit of quantification (LOQ).

Although TTX had previously been detected in *D. hystrix*, this is the first time that this toxin, distributed throughout the different organs of the fish, has been found in specimens collected in El Salvador. Hence, the consumption of *D. hystrix* collected in the region constitutes a potential threat of food poisoning. The traces of TTX detected in specimens collected in Salvadoran waters contrast with the quantifiable amounts reported from specimens collected on Malaysian, South East Indian, and Mozambican coasts and with nondetection in Hawaiian and Okinawan waters. *D. hystrix* represents another case where the concentration of TTX depends on the place of collection.

Extending the methodology proposed here to specimens collected on other coasts of El Salvador would contribute to mapping the prevalence of TTX in the country. We encourage further research into both TTX quantification and the geographical distribution of *D. hystrix* and other pufferfishes in Salvadoran waters.

## 4. Materials and Methods

### 4.1. General

All the reagents used for sample preparation, desalting, and solvents for MS analysis were of the highest quality available. Reference tetrodotoxin (TTX) material was supplied through a collaboration with the Spanish Company Laboratorios Dr. Esteve S.A. (Barcelona, Spain). The certificate of analysis of the supplied sample showed a purity of 97.1%.

### 4.2. Specimen Collection

Specimens of female (3), male (1), and undefined gender (2) of *D. hystrix* were collected by artisanal fishermen using gillnets on 1 October 2014 at Punta Chiquirín, Department of La Unión, El Salvador (13°17′31.01″ N, 87°47′03.60″ W) (Figure 5).

A male specimen of *D. hystrix* is shown in Figure 6 (a female specimen is displayed in Appendix A). The specimens were grouped by gender and washed with deionized water.

Their organs were dissected to obtain sample tissues of the liver, muscle, gonads, kidneys, and skin (Table 1, Appendix A). The tissue samples were promptly frozen and transported to Universidade da Coruña-Spain for subsequent processing and analysis.

### 4.3. Sample Preparation

Sample preparations were carried out through the methodology shown in Figure 7. After defrosting the organs, the tissues were extracted twice with 40 mL of CH_3_OH/H_2_O 1:4 (1% CH_3_COOH) in an ultrasonic bath at 55 Hz (5 min) and then centrifuged at 10,000 rpm for 30 min. The organic solvent was evaporated from the supernatant under reduced pressure to afford an aqueous phase (aqueous fraction 1) that was extracted twice with 25 mL of CH_2_Cl_2_. After removing the CH_2_Cl_2_ under reduced pressure, the resulting aqueous fraction (aqueous fraction 2) was ultra-filtered (Amilcon^®^ system) with two membrane filters of 100,000 Da and 10,000 Da, and the filtrate was submitted to fine Bio Gel P-2 column chromatography (pH = 5.5) for desalting (Appendix A). The Bio Gel P-2 column was eluted first with H_2_O to remove the salts and then with 0.5 mL of AcOH (0.03 mol/L) to obtain a toxin-enriched fraction (Fraction G). Finally, the samples were lyophilized before undergoing MS analyses [50].

### 4.4. Methodology

The stock solution was prepared by dissolving the TTX standard in CH_3_COOH (1 mM). Then, a 9-point calibration curve was prepared by serial dilution, covering a TTX concentration range of 6.3–1010 ng/mL (Appendix A, Appendix A). The solution matrix was kept at −20 °C. The detection of the toxin in the *D. hystrix* solutions was tested by using HPLC-HRMS/MS-SRM, HPLC-HRFTMS, and HPLC-HRFTMS^2^ by an in-house method. The corroboration of the HPLC-MS/MS method was based on the estimated limit of quantification (LOQ) and transition ion ratio (TIR) tolerance, as well as on the retention time tolerance. The corroboration of the HRMS method was focused on the reproducibility of retention time and on the detection or absence of ^13^C_1_/^12^C_n−1_ ions ratio in both the standard and several blank solutions. The HRMS calculations were undertaken using five significant figures (Sig. figs.), indicated in subscript format, although only four were used.

### 4.5. Analytic Methodologies Used in Tetrodotoxin Detection

#### 4.5.1. HPLC-MS/MS-SRM

Measurements were performed utilizing an HPLC-MS/MS API 3200 System (Applied Biosystems) supplied with a triple quadrupole MS/MS. The retention time, *m*/*z,* and presence of parent-product ion transitions were taken into account for a positive identification [51,52]. The following analytical conditions were employed: Mobile phase: CH_3_CO_2_NH_4_ 16 mM/CH_3_CN (3:7), pH 5.5, 40.0 min isocratic elution in a TSKgel Amide-80 HILIC-column (150 × 2.0 mm i.d.; 5 µm, Tosho, Tokyo Japan), flow 0.2 mL/min, column oven temperature: 25 °C. Sample volume injection: 5 µsL. Two ion transitions: 320 → 302 and 320 → 162, corresponding to [M + H-H_2_O]^+^ and [C_8_H_8_N_3_O]^+^ ions of TTX fragmentation, respectively, were used. Ionization was achieved via electrospray ionization (ESI) in positive mode with selected reaction monitoring (SRM).

The occurrence of the toxin was determined by comparing the retention time variation (Δ*t_R_*) with the transition ion ratio (TIR) of the standard variables [53,54]. They were computed using the following equation:(1)$${∆t}_{R}=\left|t_{R,TTX}-t_{R,sample}\right|$$
(2)TIR=∑i=1nArea1,iArea2,in
where *t_R,TTX_* is the retention time of the *TTX* standard sample and *t_R,sample_* is the retention time of any of the *D. hystrix* samples; *Area*_1,*i*_ and *Area_2,i_* correspond to the area of 320/302 and 320/162 transitions, respectively, at each concentration of the calibration curve, *n* being the number of points of the curve.

Preparation of the calibration curve: A concentrated TTX solution was prepared gravimetrically and utilized on a daily basis, ensuring freshness. Nine calibration standards were prepared from that solution by serial dilution at 6.3, 13, 25, 50, 100, 200, 600, 800, and 1010 ng/mL and used to plot the calibration curve (*R*^2^_(320→302)_ = 0.996; *R*^2^_(320→162)_ = 0.994; LOQ: 320 ng/mL; LOD: 76 ng/mL). The Δ*t_R_* tolerance used was 0.1 min [55], and the TIR tolerance was obtained as 1.3379 ± 0.2006 (TIR ± 15%). The calibration curve and the parameters derived from the calibration process are presented in the Appendix A.

#### 4.5.2. HPLC-HRFTMS

Acquisition parameters: They were set to increase analyte response and diminish isobaric ion interference as much as possible. HPLC-HRFTMS analyses were accomplished in an LTQ-Orbitrap Discovery mass spectrometer coupled with a Thermo Scientific Accela ESI-HPLC system. The chromatographic separation was achieved using a TSKgel Amide-80 HILIC-column (150 × 2.0 mm i.d., 5 µm, Tosho, Tokyo, Japan) at 25 °C and a 40.0 min isocratic elution with a mobile phase of CH_3_CO_2_NH_4_ 16 mM/CH_3_CN (3:7) pH 5.5. The sample was dissolved in CH_3_COOH (1%). Injection volume: 25 µL. Elution flow: 0.2 mL/min. Scan Event: FTMS. Mode: Full scan: 65–450. Resolution: 30,000. ESI-(+) ion source parameters were set as follows: Capillary temp: 350 °C. Capillary voltage: 26.00, and Source voltage: 4.50 kV.

Data analysis: The accurate mass measured (*m*) was calculated by averaging the masses detected (*m_i_*) at three different randomly selected peak positions in the extracted ion chromatogram (Figure 2). The mass measurement error (or accuracy) was determined by calculating the difference in ppm between the theoretical exact mass (*m_a_*) and *m* (∆*m*/*z*) [56,57] and expressed as ^12^C_n_ (^13^C_1_^12^C_n−1_) ppm.
(3)$$m=\sum_{i=1}^{3}\frac{m_{i}}{3}$$
(4)$$∆m/z=\left|\frac{m_{a}-m}{m_{a}}\times 10^{6}\mathrm{ppm}\right|$$

The natural isotope pattern was also included as an additional means to refine the potential candidates for toxin identification, and it is presented as Relative Isotope Abundance (*RIA*) of ^13^C_1_^12^C_n−1_ relative to ^12^C (5) [56,58]. The theoretical (*RIA_Theo_*) and the experimental *RIA* (*RIA_exp_*) were automatically computed using ChemDraw V 20.1 and Thermo Xcalibur V 3.0, respectively. The identification was deemed valid when the isotopic ion abundance ratio error (*RIA_error_*) remained below the maximum recommended value of ǀ16%ǀ for positive ionization mode [ESI(+)-full scan] on peaks exhibiting intensities ranging between 1 × 10^5^ and 1 × 10^6^ [58,59].
(5)$$RIAerror\left(\%\right)=\left|\frac{{RIA}_{exp}-{RIA}_{theo}}{{RIA}_{theo}}\right|\times 100$$

#### 4.5.3. HPLC-HRFTMS^2^

Sample preparation: Samples were concentrated by manually submitting them through a Waters Sep-Pak Plus C_18_ Short Cartridge (Part No. WAT020515) to mitigate any possible matrix interferences (Figure 7). Analysis conditions: The equipment and separation conditions were the same as in the HPLC-HRMS section, although the scan type was in Selected Reaction Monitoring Mode (SRM). The MS conditions were: parent ion mass (*m*/*z*) 320.00, Normalized Collision Energy (eV): 35.0, Acquisition Time (ms): 30.00, and mass range detected (*m*/*z*): 320.10→301.50–302.50 and 320.10→161.50–162.50, corresponding to [M + H-H_2_O]^+^ and [C_8_H_8_N_3_O]^+^ ions, respectively. Resolution: 30,000 and CID: 35%. Data analysis: Molecular formulae and Ring/Double Bond Equivalents (DBE) were computed in Xcalibur V 3.0 by using the exact mass detected. Fragments selected for the analysis met two criteria: they were abundant during fragmentation and did not originate from the same part of the molecule under study.

### 4.6. Post-Acquisition Data Processing

The thermo Orbitrap files (*.Raw) were handled in Xcalibur V3.0 from Thermo Fisher Scientific V3.0 and MZmine V 2.53 [49,60]. The HPLC-HRMS^2^ data analyses were carried out as previously described [61,62]. The calibration curve, the linear regression analysis, and the graphics were carried out using Microsoft Excel (Microsoft 365 MSO, version 2309). The chemical structures were performed in ChemDraw V20.1. The figures were handled in Affinity Designer V 2.1. The treatment of MZmine data was carried out, as shown in Figure 8. For a detailed software setup, see SM.

## Figures and Tables

**Figure 1 toxins-15-00409-f001:**
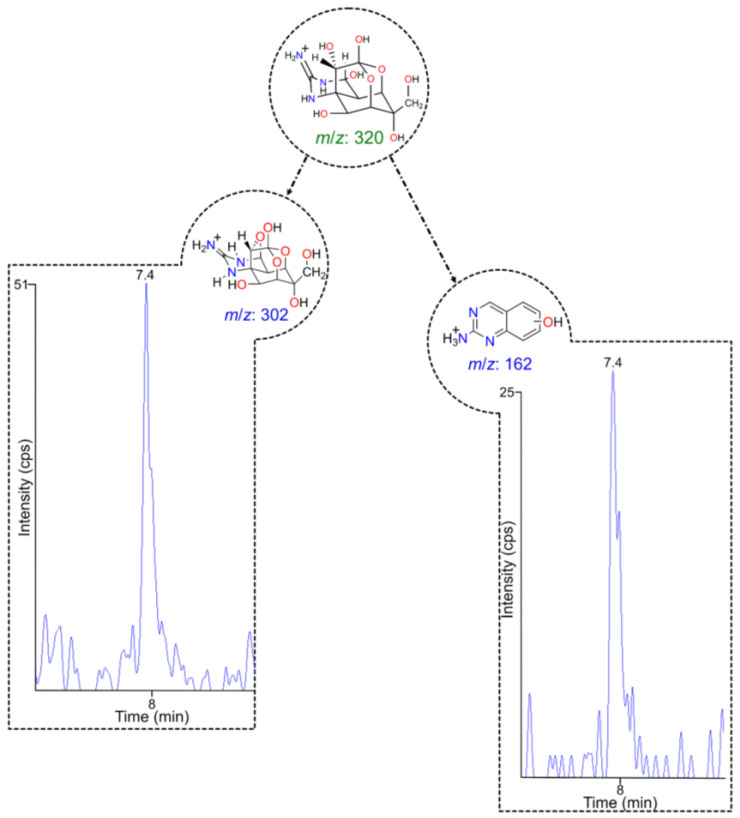
Selected reaction monitoring (SRM) chromatograms of female gonads (FG) samples, where the parent ion (*m*/*z* 320) is fragmented into the product ions (*m*/*z* 302 and 162). Δt*_R_* found was 0.03 min, and TIR found was 0.59.

**Figure 2 toxins-15-00409-f002:**
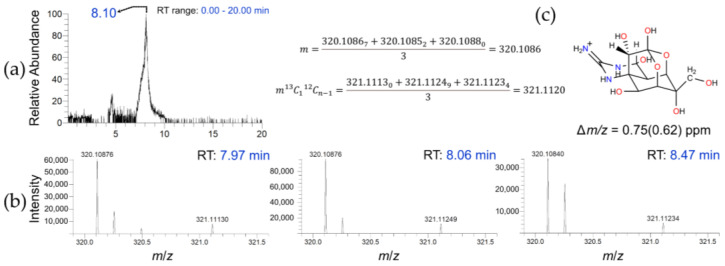
HPLC–ESI(+)–HRFTMS (SIM mode) analysis of female skin (FS) sample of *D. hystrix*. Extracted ion chromatogram, mass range: *m*/*z* 320.0–320.20 (**a**) and high-resolution mass spectra at different retention times (**b**)-row. Occurrence of TTX in the FS sample was assured by the accuracy in the detected mass (Δ*m*/*z* = 0.75 (0.62) ppm). The calculation model is also shown (**c**).

**Figure 3 toxins-15-00409-f003:**
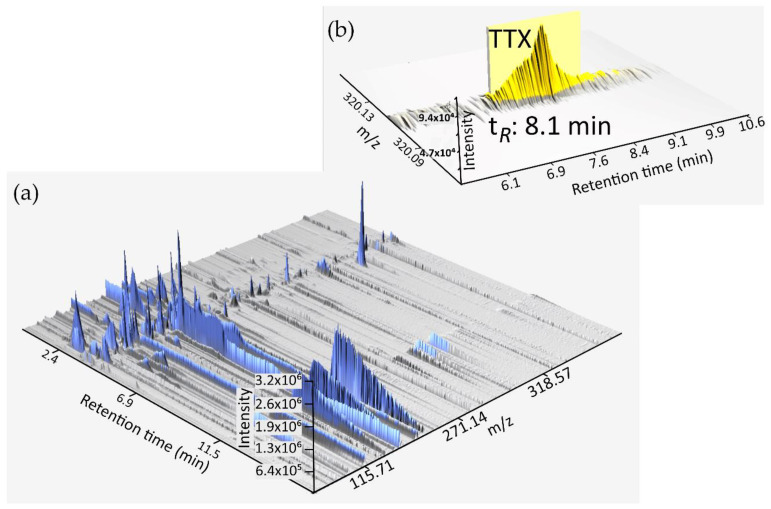
HPLC–HRFTMS 3D projection of *D. hystrix*-FS sample generated after data processing with MZmine (**a**). Extracted ion chromatogram (marked yellow) showing the peak attributed to TTX (**b**).

**Figure 4 toxins-15-00409-f004:**
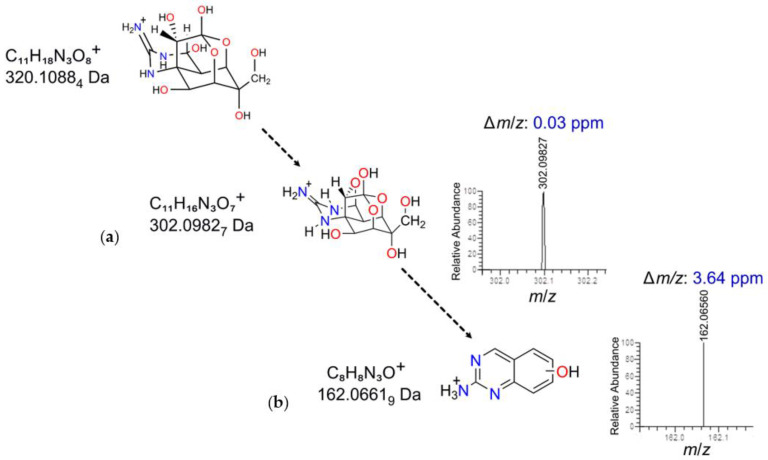
TTX was identified in the LU sample of *D. hystrix* via a LTQ-Orbitrap XL and by the fragments generated during MS^2^ analysis. Fragmentation of the ion corresponding to the compound eluted at 28.7 ± 0.1 min showing the distinct fragment at *m*/*z* 302.0983 Da (Δ*m*/*z* 0.03 ppm) attributed to the loss of a water molecule in TTX (**a**). Second fragmentation showing the ion at *m*/*z* 162.0656 Da (Δ*m*/*z* 3.64 ppm) attributed to the characteristic 2-aminohydroxyquinazoline fragmention detected within the TTX fragmentation analysis (**b**).

**Figure 5 toxins-15-00409-f005:**
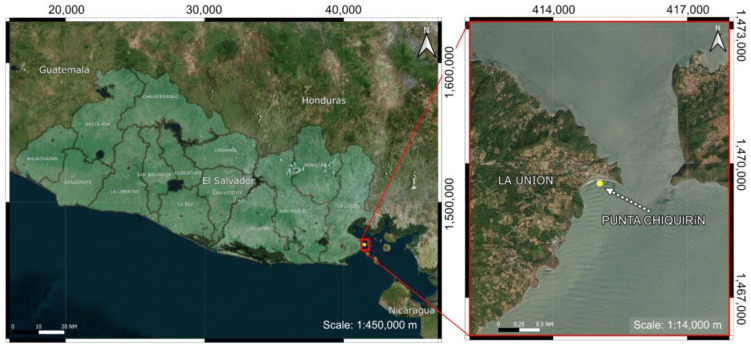
Sample collection place: Punta Chiquirín, Department of La Unión, Republic of El Salvador.

**Figure 6 toxins-15-00409-f006:**
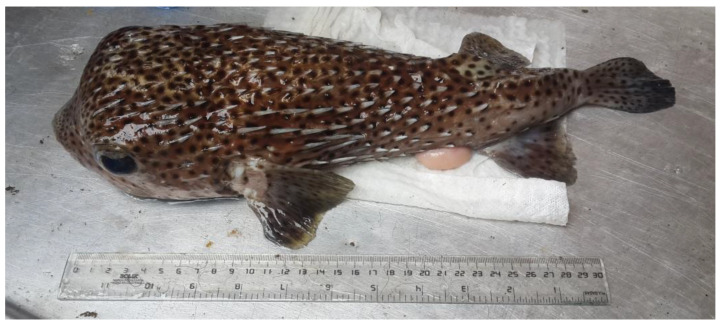
Male specimen of *D. hystrix* collected in Punta Chiquirín, Republic of El Salvador. Scale shown in centimeters.

**Figure 7 toxins-15-00409-f007:**
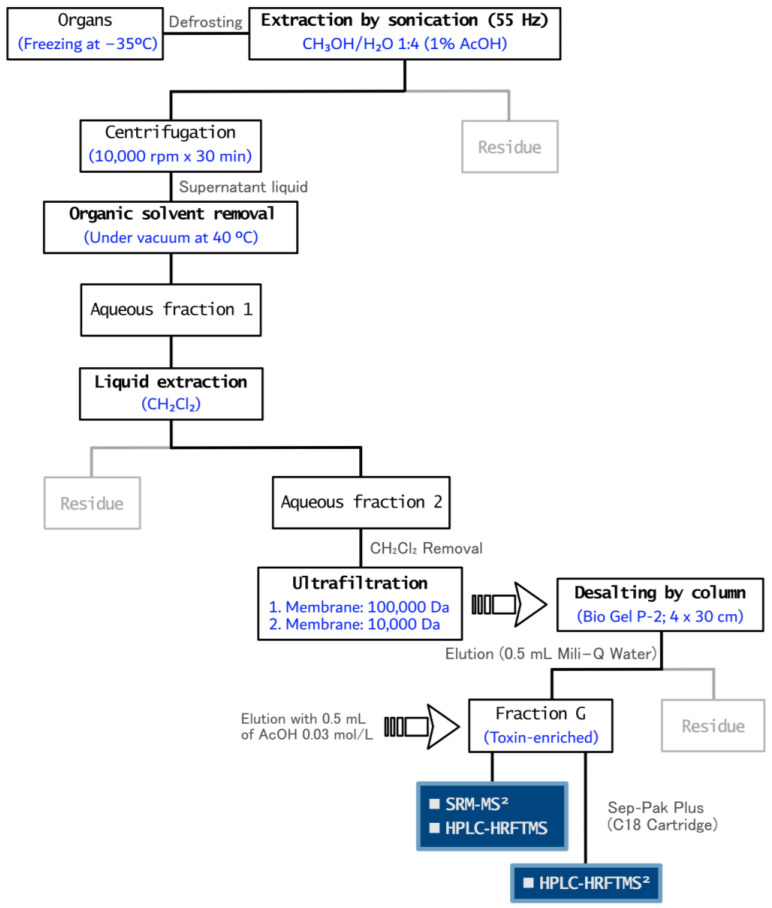
Flowchart employed for TTX detection in *D. hystrix*.

**Figure 8 toxins-15-00409-f008:**
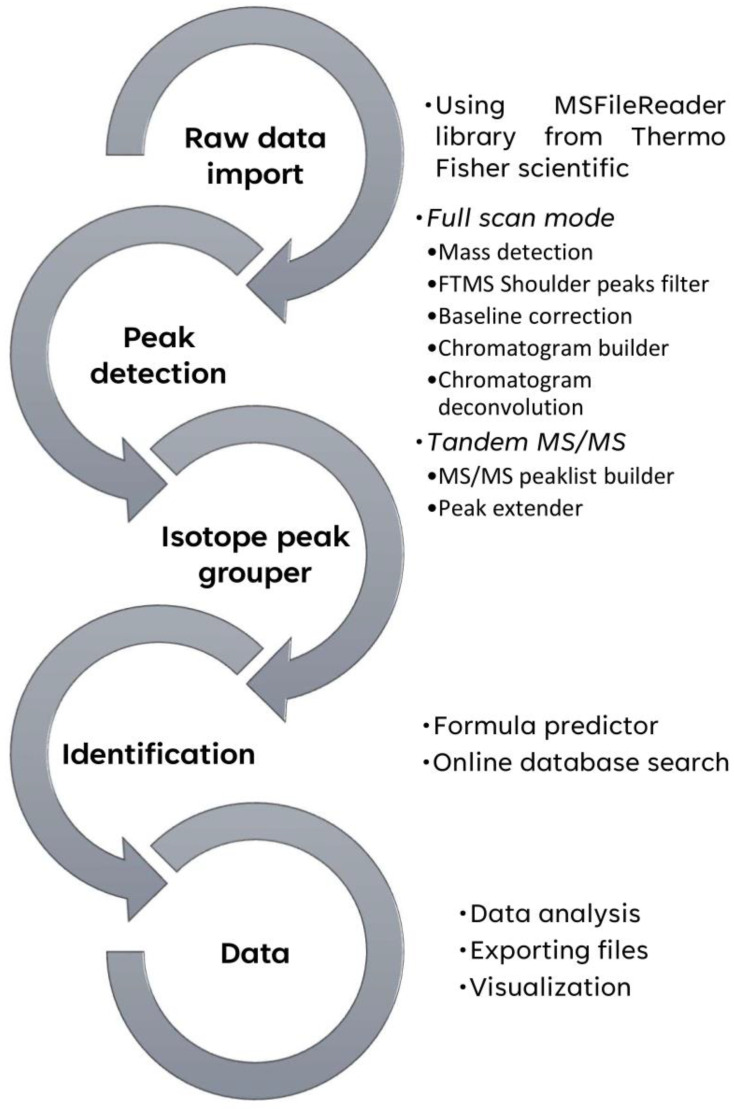
MZmine data analysis workflow for HPLC-HRFTMS files processing/dereplication of TTX.

**Table 1 toxins-15-00409-t001:** Physical characteristics and extract weights of the *D. hystrix* specimens studied.

Gender	Weight (g) ^a^	L. (mm) ^b^	Weight ^c^	Liver	Gonads ^d^	Kidney	Skin	Muscle
Female	913.9	199.3 ± 4.5	Organs (g)	82.9	17.7	8.5	87.2	95.4
			Raw Extr. (g)	10.0	7.6	3.4	10.0	10.0
			Ultrafiltr. (mg)	65.4	156.3	400.8	45.3	227.6
Male	1177.3	406.0	Organs (g)	33.5	6.5	10.1	22.5	75.3
			Raw Extr. (g)	10.0	4.0	4.8	10.0	10.0
			Ultrafiltr. (mg)	160.4	29.9	52.2	39.9	91.4
Undef.	411.9	184.0 ± 3.7	Organs (g)	39.8	3.5	3.2	52.6	48.6
			Raw Extr. (g)	10.0	1.6	1.0	10.0	10.0
			Ultrafiltr. (mg)	75.3	9.7	17.2	143.2	125.2

Undefined gender (Undef.); Raw extracts (Raw Extr.): weight of dried acidulated methanol extracts; Ultrafiltration (Ultrafiltr.): weight of extract obtained after ultrafiltration procedure. ^a^ These amounts correspond to the sum of the weights of three females, one male, and two undefined gender specimens, respectively; ^b^ Specimen length (L) is expressed as the average of collected specimens; ^c^ These amounts correspond to the combination of the organ weights and subsequent Raw Extr., and Ultrafiltr. weights of each gender; ^d^ Gonads are referred to as testicles (Male), ovaries (Females), and gonads (Undefined gender).

**Table 2 toxins-15-00409-t002:** Summary of Tetrodotoxin detection in extracts of *D. hystrix*.

Gnd. Spec.	Organs				
	Liver	Gonads	Kidney	Skin	Muscle
Female	-	HPLC-MS/MS-SRM *	-	HPLC-HRFTMS	-
Male	HPLC-HRFTMS	-	-	-	-
Undef.	HPLC-HRFTMS^2^	-	-	-	HPLC-MS/MS-SRM *

Gnd. Spec.: Gender specimen. Undef.: Undefined (-): Not detected. (*) The presence of TTX was not confirmed because the transition ion ratio (TIR) was outside the acceptable range for this technique.

## Data Availability

All data can be obtained from J.R. or C.J. at CICA at Universidade da Coruña.

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
