# Peer review of "Multistage Detection of Tetrodotoxin Traces in Diodon hystrix Collected in El Salvador"

_toxins, 2023, doi:10.3390/toxins15070409_

Round 1

Reviewer 1 Report

1.        The SRM method only detected trace amounts of TTX and did not allow for accurate quantitative analysis. The Bio Gel P-2 column chromatography was used in the sample processing, which may have resulted in low recovery of TTX. In addition, the detection limits and sensitivity of the SRM method and the HRFT full scan method are not known.

 2.        In this study, full scan and MS2 analyses were performed separately, so why not put these two scans in one method, which would be simple and fast.

 3.        The retention time for TTX was at 8 min, why was the liquid phase method run for 40 min?

 4.       In the Introduction, the current advances and problems in the detection of TTX in fish by LC-MS should be described.

 5.        There are several errors of “Error! Reference source not found” in the text.

 6.        “progenitor ion” is suggested revised to “parent ion” or “precursor ion”.

 7.        Table 1, Male fish were larger in size and weight than female fish, but why were the weights of different tissues in males almost always smaller than in females?

 8.        Fig. S5, the intensity of the signal response was too low in the fish sample, where 320/162 were probably below the LOD.

 9.        Fig. 3b, why 3 different retention times appear? The m/z can be displayed for a period of retention time.

Author Response

Thank you very for the reviewer’s comments.

Comments to the Author

  1. The SRM method only detected trace amounts of TTX and did not allow for accurate quantitative analysis. The Bio Gel P-2 column chromatography was used in the sample processing, which may have resulted in low recovery of TTX. In addition, the detection limits and sensitivity of the SRM method and the HRFT full scan method are not known.

Answer to reviewer 1:

Thank you very for the reviewer’s comments. We agree with the reviewer in relation to Bio Gel P-2 column chromatography may have resulted in low recovery of TTX but it has been used for a long time as a common step in the separation process.

In relation to the detection limits and sensitivity of the used methods:

For the SRM method, LOD (0.076 µg mL-1) and LOQ (0.32 µg mL-1) were computed using equations 1 and 2, respectively following the methodology described in Ref. Miller, J.N.; Miller, J.C.; Miller, R.D. Statistics and chemometrics for analytical chemistry, Seventh edition ed.; Pearson Education Limited: Harlow, United Kingdom, 2018. The procedure is now described in the SM. Sensitivity was 1301580.75.

For LC-HRMS method: LOD, LOQ and sensitivity were not estimated since we did not build up a calibration curve.  Detection was assured by: retention time, HRMS (12Cn, and 12Cn-113C1). HRMS and RIAerror were within the known tolerance described in literature to the Orbitrap detector. We even commented on the Obitrap limitation related to the “lower threshold setting, below which no ions can be detected or accurately reported”.

  1. In this study, full scan and MS2 analyses were performed separately, so why not put these two scans in one method, which would be simple and fast.

Answer to reviewer 1: Although the reviewer is right in suggesting the combination of both methods, we decided to run the experiments separately because we wanted to see the ability of each method. Some labs do not have MS2 equipment, and we wanted to test if full scan was enough in this case. 

  1. The retention time for TTX was at 8 min, why was the liquid phase method run for 40 min?

Answer to reviewer 1: We run for 40 min because we wanted to make sure that other TTX analogues were no present.

  1. In the Introduction, the current advances, and problems in the detection of TTX in fish by LC-MS should be described

Answer to reviewer 1: A very recent reference describing the current advances and limitations in the detection of TTX by LC-MS was included in the introduction following the reviewer´s comment.

  1. There are several errors of “Error! Reference source not found” in the text

Answer to reviewer 1: The “Error! Reference source not found” in the text were removed.

  1. “progenitor ion” is suggested revised to “parent ion” or “precursor ion”.

Answer to reviewer 1: “progenitor ion” was replaced by “parent ion” through the manuscript following the reviewer’s comment.

  1. “Table 1, Male fish were larger in size and weight than female fish, but why were the weights of different tissues in males almost always smaller than in females?

Answer to reviewer 1: The weights correspond to the sum of the specimens for gender: three females, one male, and two undefined. Because this information was no clear, we modified Table 1 to clarify this issue and other definitions in the legend.

  1. Fig. S5, the intensity of the signal response was too low in the fish sample, where 320/162 were probably below the LOD

Answer to reviewer 1: The reviewer is right. This fact is shown in the legend of this figure: “The expected transition ion ratio (TIR) between the two quantifications ions was not found in the sample, presumably due low sample concentration”.

  1. Fig. 3b, why 3 different retention times appear? The m/z can be displayed for a period of retention time

Answer to reviewer 1:

The m/z of TTX is displayed at three retention times following the original article by Macarov (ref. 37 and 38): the accurate mass measured is computed as the average of the mass detected at three different randomly selected peak positions in the extracted ion chromatogram.

Reviewer 2 Report

The manuscript under reviewing provides detection of Tetrodotoxin traces in Diodon hystrix. In the current version, the MS cannot be published in the Toxins journal. See below for my major and minor concerns. Unfortunately, I am not specialist on MS technology, thus my concerns only about biology of TTX-bearing animals.

Major concerns

The MS does not provide any useful scientific information about the toxicity of fish Diodon hystrix. Thus, the authors indicate the presence of TTX in certain organs of Diodon hystrix, but do not provide any data on the concentration of the toxin or the presence other TTX analogues. Based on obtained data the authors put forward hypotheses that have long been put forward before, and its does not have any scientific novelty:

1.     (Line 205-206) – «A trace detection methodology by MS techniques was proposed and applied to detect TTX in specimens» The authors used three different MS techniques to find the toxin in fish which is already known to contain the toxin, i.e. there is no novelty.

2.     (Line 218-219) – «presence of TTX in D. hystrix is associated with its geographic distribution» For many puffer fish, it has been shown that the concentration of TTX depends on the place of catch.

English very difficult to understand.

Minor concerns

Line 22: “  Since its discovery in 1964 by Y. Tahara” – Do You mean Yoshizumi Tahara, who, first discovered in 1909 and extracted the toxin from the ovaries.

Line 24-25 “TTX is distributed intra-organismically at almost all levels in” – sounds ugly, please, rewrite this statement.

Line 26-28 – This is wrong statement. Primary TTX is synthesized by microorganisms and then toxin migrated through food chain, accumulated in predator animals (https://doi.org/10.1016/j.cbd.2005.11.003.)

Line 29-33: It is very ordinary information – I recommend to remove this paragraph.

Line 82: “D. hystrix” must be italic. Please, verify and correct all over the MS.

Line 116-117. Very strange statement. Authors described “Limit of detection”. Please, remove or rephrase this statement.

Table 6: Giving a google map without any processing is extremely bad idea. Authors have to remove goole icons at least.

Author Response

Thank you very for the reviewer’s comments.

Comments to the Author

The MS does not provide any useful scientific information about the toxicity of fish Diodon hystrix. Thus, the authors indicate the presence of TTX in certain organs of Diodon hystrix, but do not provide any data on the concentration of the toxin or the presence other TTX analogues. Based on obtained data the authors put forward hypotheses that have long been put forward before, and its does not have any scientific novelty:

  1. (Line 205-206) – «A trace detection methodology by MS techniques was proposed and applied to detect TTX in specimens» The authors used three different MS techniques to find the toxin in fish which is already known to contain the toxin, i.e. there is no novelty.

Answer to reviewer 2: This work constitutes the first study on the presence of TTX in fish specimens collected in El Salvador. We applied a sensitive three-stage approach to detect TTX traces in marine organisms using MS experiments by a combination of HPLC-HRMS, HPLC-MS/MS-SRM, and HPLC-HRMS2 techniques. Furthermore, the trace amount of TTX detected in D. hystrix collected in El Salvador contrast with other studies of specimens collected in other parts of world, confirming the proposal that its presence in D. hystrix is associated with its geographic distribution. Any TTX analogue was not detected in this study.

  1. Line 218-219) – «presence of TTX in D. hystrix is associated with its geographic distribution» For many puffer fish, it has been shown that the concentration of TTX depends on the place of catch.. English very difficult to understand

Answer to reviewer 2: To clarify this part, we have replaced for: “The traces of TTX detected in specimens collected in El Salvador contrast with the quantifiable amounts reported from specimens collected in Malaysian, South East Indian, and Mozambican coasts, and its non-detection in Hawaiian and Okinawan waters. D. hystrix represents another case where the concentration of TTX depends on the place of collection.”

  1. Line 24-25 “TTX is distributed intra-organismically at almost all levels in” – sounds ugly, please, rewrite this statement

Answer to reviewer 2: This sentence was replaced by “Although this toxin has been found in several organs of adult pufferfish, TTX is mostly localized in skin, muscles, intestines, gonads, kidneys, and liver “

  1. Line 26-28 – This is wrong statement. Primary TTX is synthesized by microorganisms and then toxin migrated through food chain, accumulated in predator animals (https://doi.org/10.1016/j.cbd.2005.11.003.)

Answer to reviewer 2: This statement was changed in the manuscript and replaced by “The true origin of TTX remains controversial. Its production by symbiotic bacteria (endogenous route) or by accumulation through the diet are the most accepted proposals. Moreover, feeding of TTX-bearing microalgae by fish and their consequent accumulation is another explanation”.

  1. Line 22: “ Since its discovery in 1964 by Y. Tahara” – Do You mean Yoshizumi Tahara, who, first discovered in 1909 and extracted the toxin from the ovaries

Answer to reviewer 2: The reviewer is right. Tahara and Hirata reported the first record of this toxin from the ovaries of fish at the beginning of the 20th century. Consequently, we modified this sentence as:

“Since the first report of this toxin isolated from the ovaries of a puffer fish by Tahara and Hirata at beginning of the 20th century, …

  1. Line 82: “D. hystrix” must be italic. Please, verify and correct all over the MS.

Answer to reviewer 2: D. hystrix was written in italic.

  1. Line 116-117. Very strange statement. Authors described “Limit of detection”. Please, remove or rephrase this statement.

Answer to reviewer 2: We rephrased this statement: “The presence of TTX was not confirmed due to…”.

  1. Table 6: Giving a google map without any processing is extremely bad idea. Authors have to remove google icons at least

.Answer to reviewer 2: The map was corrected following the reviewer´s comment.

Reviewer 3 Report

The manuscript “Multistage detection of Tetrodotoxin traces in Diodon hystrix collected in El Salvador” describes a methodology to detect trace levels of TTX in the porcupine pufferfish. This is of great interest, given that the presence of TTX in this species shows high variability among different regions. Therefore, the method described here will contribute to a better knowledge of toxin accumulation and distribution in this group of fish.  

The manuscript is clear and the figures and tables provided in the article and supplementary material allows for a good understanding of the work process and results.

Minor remarks:

Throughout the text: check italics in Latin names

Table 1.

Specimen length (L), is expressed as the average of collected specimens”. Add deviation.

“Weight is expressed in mg.” in the table is written “g”.

Line 221. Include the origin of the commercial standard.

Line 224. Could you provide details about how the fishes were collected?

Line 235. Place the citation at the end of the phrase.

Figure 6. The map could be improved. The sampling site is not clear.  

Author Response

Thank you very for the reviewer’s comments.

Comments to the Author

  1. The manuscript “Multistage detection of Tetrodotoxin traces in Diodon hystrix collected in El Salvador” describes a methodology to detect trace levels of TTX in the porcupine pufferfish. This is of great interest, given that the presence of TTX in this species shows high variability among different regions. Therefore, the method described here will contribute to a better knowledge of toxin accumulation and distribution in this group of fish.

The manuscript is clear and the figures and tables provided in the article and supplementary material allows for a good understanding of the work process and results..  .

Answer to reviewer 3: Thank you for your comments.

  1. Throughout the text: check italics in Latin names

Answer to reviewer 3: All the Latin names throughout the text were checked and written with italics

  1. Table 1. “Specimen length (L),is expressed as the average of collected specimens”. Add deviation.

“Weight is expressed in mg.” in the table is written “g”.

Answer to reviewer 3: The deviation of the specimen length (L) for female and undefined was added in Table 1 because there is just one male specimen. The weight expressions were clarified in Table 1.

  1. Line 221. Include the origin of the commercial standard.

Answer to reviewer 3: TTX sample was supplied through a collaboration with the Spanish Company Laboratorios del Dr. Esteve S.A.

  1. Line 224. Could you provide details about how the fishes were collected?.

Answer to reviewer 3: The puffer fish specimens were collected by artisanal fishermen using gillnets. This information was enclosed in Material and Methods.

  1. Figure 6. The map could be improved. The sampling site is not clear.

Answer to reviewer 3:  The map of Figure 6 was improved, and the sample collection places were indicated.

Round 2

Reviewer 1 Report

The manuscript has been revised in detail in the light of the reviewers' comments. It is suggested that the manuscript be accepted.

Author Response

Thank you very for the reviewer’s comments.

Reviewer 2 Report

After the correction, the authors did not improof the article. The main problem of the MS is the lack of novelty. The authors used routine chromatography methods, which are widely used, among other things, for the detection of toxins. The fact that the authors detected minute amounts of TTX in Diodon hystrix is also not new. This species has long been known for containing TTX. My recommendations for improving the MS ("provide any data on the concentration of the toxin or the presence of other TTX analogues") were not made by the authors. Article must be rejected

Author Response

Thank you very for the reviewer’s comments.

Comments to the Author

After the correction, the authors did not improof the article. The main problem of the MS is the lack of novelty. The authors used routine chromatography methods, which are widely used, among other things, for the detection of toxins. The fact that the authors detected minute amounts of TTX in Diodon hystrix is also not new. This species has long been known for containing TTX.

Answer to reviewer 2: We had introduced in the revised version a lot of modifications following the suggestions of the reviewers to improve the article. Although we have used known chromatography methods, the combination of the three-stage approach to detect TTX traces is not common. On the other hand, this work represents the first study on the presence of TTX in fish specimens collected in El Salvador and gives more insights about its presence in D. hystrix from fish collected in different places.

My recommendations for improving the MS ("provide any data on the concentration of the toxin or the presence of other TTX analogues") were not made by the authors. Article must be rejected.

Answer to reviewer 2: Thank you very for the recommendation.

Although the exact concentration of the toxin is not possible to measure, it is possible to give a value within a range. Thus, we modified the manuscript in the section 2.1.3. HPLC-MS/MS-SRM by including this sentence:

“These findings indicate that the toxin concentration in FG and MU would be between the limit of detection (LOD: 72 ng/mL) and the limit of quantification (LOQ: 320 ng/mL). See the calculation of detection and quantification limits of HPLC-MS/MS-SRM in Figure S4 in the SM. Thus, the concentration of TTX in those organs would be within the range of 4.1 and 18.1 μg.Kg-1 which would equivalent to a value between 2 and 9 Mouse units.

On the other hand, we have performed the search of the most common TTX analogues by using HPLC-HRFTMS in the FS and ML samples and we did not detected any of them. We have included this information in the revised manuscript by including a new sentence:

“Once TTX was detected in the FS and ML samples by using the HPLC-HRFTMS method, we endeavored to find the most common TTX derivatives (4,9-anhydro-TTX; 5-deoxy-TTX; 11-deoxy-TTX; 5,11-dideoxy-TT; 6,11-dideoxy-TTX; 5,6,11-trideoxy-TTX 11-nor-TTX-6(R)-ol; and 11-oxo-TTX) in those samples by resorting to that methodology.[43,44] As a result, no other analogue of TTX was detected in either sample”.